

# On the exponent in the Von Bertalanffy growth model

Katharina Renner-Martin, Norbert Brunner[*], Manfred Kühleitner[*], Werner Georg Nowak[*] and Klaus Scheicher

Department of Integrative Biology and Biodiversity, Institute of Mathematics, Universität für Bodenkultur Wien, Vienna, Austria

[*] These authors contributed equally to this work.

## ABSTRACT

Von Bertalanffy proposed the differential equation $m'(t) = p \times m(t)^a - q \times m(t)$ for the description of the mass growth of animals as a function $m(t)$ of time $t$. He suggested that the solution using the metabolic scaling exponent $a = 2/3$ (Von Bertalanffy growth function VBGF) would be universal for vertebrates. Several authors questioned universality, as for certain species other models would provide a better fit. This paper reconsiders this question. Based on 60 data sets from literature (37 about fish and 23 about non-fish species) it optimizes the model parameters, in particular the exponent $0 \leq a < 1$, so that the model curve achieves the best fit to the data. The main observation of the paper is the large variability in the exponent, which can vary over a very large range without affecting the fit to the data significantly, when the other parameters are also optimized. The paper explains this by differences in the data quality: variability is low for data from highly controlled experiments and high for natural data. Other deficiencies were biologically meaningless optimal parameter values or optimal parameter values attained on the boundary of the parameter region (indicating the possible need for a different model). Only 11 of the 60 data sets were free of such deficiencies and for them no universal exponent could be discerned.

## INTRODUCTION

### Growth model

Size at age is a key metric of productivity for any animal population (*MacNeil et al., 2017*) and a wide range of growth models to describe the size and also the mass of animals as a function of time has been developed. Amongst applications is stock assessments in fisheries management (*Juan-Jordá et al., 2015*) and applications in ecology, e.g., understanding outbreak dynamics (*Pratchett, 2005*). Of particular interest are models based on biological principles. This paper considers a class of such models that was developed by *Von Bertalanffy (1957)*, who formulated a differential equation of ontogenetic growth:

$$\frac{dm(t)}{dt} = p \cdot m(t)^a - q \cdot m(t). \tag{1}$$

Corresponding author
Katharina Renner-Martin,
katharina.renner-martin@boku.ac.at,
kathi.renner-martin@gmx.de

PeerJ ___________________________________________

Equation (1) aims at explaining the allocation of metabolic energy between growth and sustenance of an organism, using a metabolic scaling exponent $0 \leq a < 1$: if $m = m(t)$ is body mass (weight) at age $t$, then the body utilizes resources at a metabolic rate $(p \cdot m^a)$ for growth, except for catabolism (energy use for the operation and maintenance of existing tissue) proportional to body weight $(q \cdot m)$. The parameters $p$ and $q$ are positive constants obtained by fitting the model curve (1) to growth data.

## Is there a universal exponent?

Does the modeling of the growth of different species require different metabolic scaling exponents? The null hypothesis would state that on the contrary a certain universal exponent would suffice. Several concrete values for such an exponent have been proposed in literature, resulting in a scientific controversy about the 'true' exponent (*Isaac & Carbone, 2010*).

*Von Bertalanffy (1934)*, *Von Bertalanffy (1949)* and *Von Bertalanffy (1957)* suggested that $a = 2/3$ would describe the mass growth of vertebrates; this defines the classical Von Bertalanffy growth function (VBGF). Von Bertalanffy explained this value of the exponent by a biological reasoning: Anabolism (synthesis for growth) would be proportional to the 2/3th power of body weight, as the oxygen consumption would be proportional to surface (2/3th power of volume). This value was supported e.g., by *Banavar et al. (2002)* and *White & Seymour (2003)*.

*West, Brown & Enquist (2001)* proposed an alternative value $a = \frac{3}{4}$ of the exponent, as anabolism would relate to the number of capillaries, which in turn would be proportional to the $\frac{3}{4}$th power of the number of cells (proportional to body mass). Actually, a metabolic exponent $a = \frac{3}{4}$ had been suggested much earlier by *Kleiber (1947)*. This value was also supported by *Darveau et al. (2002)* and it is widely used in animal science.

Von Bertalanffy identified also species, where mass growth would be better described by an exponent $a = 0$. Moreover, an exponent $a_L = 0$ is widely used to describe the length growth of fish and many authors reported an excellent fit also for e.g., shellfish (*Koch et al., 2015*). For instance, the FishBase database (*Froese & Pauly, 2017*) presumes this model and lists growth parameters for 2,320 species. In addition, a search in Google Scholar identified approximately 25,000 papers related to the use of this model for fish. The use of this model in literature was also specifically surveyed for elasmobranch species, showing that it was studied twice as often as any other model (*Smart et al., 2016*).

More recent literature observed that no single exponent may be exactly correct. As *Killen, Atkinson & Glazier (2010)* and *White (2010)* observed, for different species there were different optimal exponents $a_{opt}$. Also for the same species data sets differing in environmental factors (e.g., food composition, temperature) supported different exponents; c.f. *Kimura (2008)*, *Porch (2002)*, *Quince et al. (2008)*, *Stewart et al. (2013)*, or *Yamamoto & Kao (2012)*.

There were also suggestions that the very problem of determining an optimal exponent may be ill-posed. For, as *Shi et al. (2014)* observed, for some data sets a near-optimal fit could be achieved by a wide range of exponents. As a consequence, data fitting using nonlinear regression by means of the method of least squares may be impeded by numerical instability.

## Problem of the paper

The present paper develops an approach to identify best fitting exponents for model (1) despite numerical instability and applies this to explore the variability of the exponent. To this end, the growth model (1) was applied to 60 data sets and best fit exponents together with suitable best fit parameters were determined.

In order to compare the goodness of fit across different data sets, the paper applies Akaike's information criterion: given the optimal exponent $a_{opt}$, computed for a certain data set, and a hypothesized exponent $a$ (e.g., a universal value for the exponent) the Akaike weight *prob* ($a$) is the probability that the model (1) using the hypothesized exponent $a$ is true, when compared with the optimal exponent $a_{opt}$. Thereby, given a data set, the exponent $a$ is refuted for this data set, if in comparison to $a_{opt}$ its Akaike weight is below 2.5%. Variability of the exponent is measured by the length of the interval of not-refuted exponents.

# MATERIALS AND METHODS

## Choice of the growth model

Model (1) is believed to have a biological meaning (see the 'Introduction'). This was the main reason, why the authors decided to study this model in more detail. This distinguishes model (1) from simpler models recommended in literature for data interpolation, such as power-laws between size and age (*Katsanevakis & Maravelias, 2008*).

Another reason to study model (1) was the boundedness of the model function and its sigmoid shape, if $a, p, q > 0$: the rate of mass growth increases, as size increases, until it reaches a maximal value and then decreases towards zero as mass approaches the asymptotic weight limit $m_{max}$ (mature body mass). This is a unimodal curve, peaking above the weight at the inflection point. To demonstrate the plausibility of this model, Fig. 1 compares this theoretical pattern of the growth rate, i.e., the right hand side of Eq. (1), with observed growth rates from data.

As follows from this shape of the model curve for $a, p, q > 0$, $m_{max} = (p/q)^{1/(1-a)}$ is the positive zero of the right hand side of Eq. (1); the rate of growth vanishes for $m = m_{max}$. The inflection point is assumed when body mass reaches the value $m_{infl} = a^{1/(1-a)} \cdot m_{max}$; the derivative of the right hand side of Eq. (1) vanishes for $m = m_{infl}$. The fraction $m_{infl}/m_{max}$ varies between just above 0% and just below $1/e = 37\%$, the limits of $a^{1/(1-a)}$ for $a \to 0$ and for $a \to 1$, respectively.

An additional reason for the selection of model (1) was the availability of an explicit solution (2) of Eq. (1) in terms of elementary functions (*Von Bertalanffy, 1957*), whence spreadsheets could be applied for data fitting:

$$\frac{m(t)}{m_{max}} = \sqrt[1-a]{1 - (1 - (\frac{m_0}{m_{max}})^{1-a}) \cdot e^{-q \cdot (1-a) \cdot t}} \qquad (2)$$

Formula (2) explains growth in terms of four parameters: the exponent $a$, the growth coefficient $k = q \cdot (1 - a)$, the initial value (neonate weight) $m_0 = m(0)$, and $m_{max}$. (The dimensions are mass in g or kg for $m$, $m_0$ and $m_{max}$, time in d, months or years for $t$, 1/time for $q$ and dimensionless for $a$). In the literature there are different parametrizations and

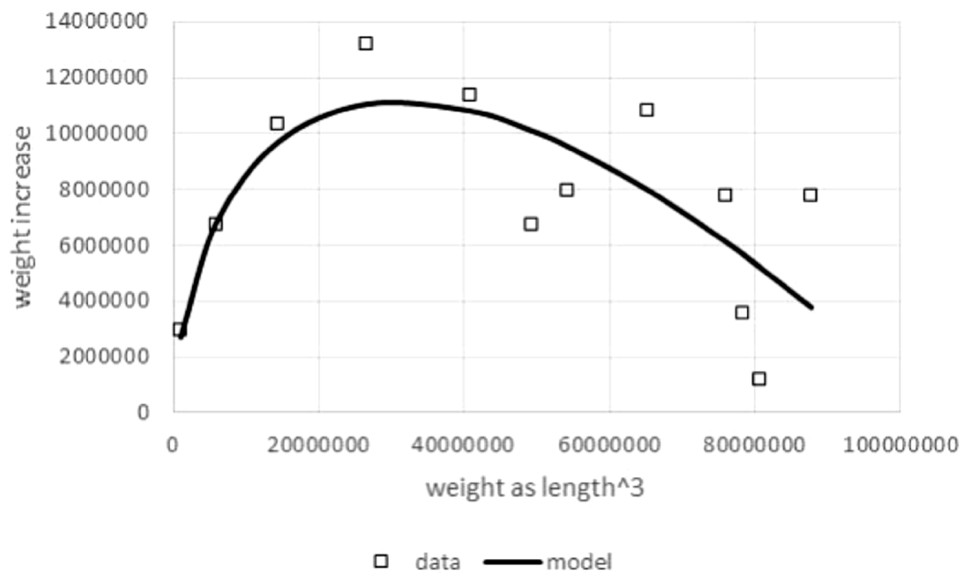

**Figure 1** **Right hand side of Eq. (1) compared to observed values for the left hand side of (1).** Figure generated in Microsoft EXCEL based on data set #10 (outlier removed). Squares indicate observed growth rates, the left hand side of Eq. (1), whereby $dm/dt$ was computed by numeric differentiation (quadratic interpolation taking care of unequal $dt$-interval length; *Burden & Faires, 1993*). The line (model) is the right hand side of Eq. (1). Its parameters ($p = 323$, $q = 0.73$) were obtained from a linear fit to the observed growth rates (LINEST function of EXCEL applied to $dm/dt$, $m^a$, and $m$ with $a = 2/3$). This method of fitting parameters is a modification of the Walford plot (*Walford, 1946*).

the present one follows a recommendation of *Cailliet et al. (2006)*. Knowing $a$ and $q$ allows to compute $m_{max}$ and $p$ from each other (using the formula for $m_{max}$): $p = q \cdot m_{max}^{1-a}$. Several papers use a time shift $t_0$ to eliminate the multiplicative constant of formula (2), but $t_0$ might not have a biological meaning (*Schnute & Fournier, 1980*).

Formula (2) is also valid for an exponent $a = 0$, where it is the model of bounded exponential growth. There, the difference $m_{max} - m(t)$ between mature body mass $m_{max}$ and current weight follows the model of exponential decay and the model curve is not sigmoid. As *Von Bertalanffy (1957)* observed, the model for length growth using the exponent $a_L = 0$ is equivalent to VBGF for mass growth with a metabolic scaling exponent $a = 2/3$. Therefore, in fishery literature both models are referred to as VBGF. However, for this paper, VBGF always means $a = 2/3$. The equivalence between the models was meant in the following sense: if mass growth is described by the VBGF with $a = 2/3$ and if mass is assumed to be proportional to the third power of length, then the growth of length is modeled by bounded exponential growth $a_L = 0$. And conversely, if length increases according to bounded exponential growth, $a_L = 0$, then mass growth follows VBGF with $a = 2/3$.

The authors acknowledge that there is room for alternative models and that certain data sets may not be appropriate for further analysis by model (1). In particular, model (1) may be generalized using more parameters. For instance, in (1) the term $q \cdot m$ for catabolism may be replaced by a more general term $q \cdot m^b$ (*Pütter, 1920*). This more general

equation includes certain well-known models, e.g., for $a = 1$ a model due to *Richards (1959)* and for $b = 2$ generalizations of the logistic growth function of *Verhulst (1838)*. However, the equation for the generalized model is no longer solvable using elementary functions (*Ohnishi, Yamakawa & Akamine, 2014*). Such more complex models may be analyzed using numerical solutions of differential equations (e.g., *Leader, 2004*), but then numerical errors would require further analysis. Further, while for such models with more free parameters the fit of the model curve to the data may be better, overfitting may result in unrealistic parameter values.

Summarizing, with four parameters model (1) appeared to be flexible enough to represent growth curves of different sigmoid shapes and to allow for handling numerical instability, when analyzing the problem of the variability of the parameters.

## Data

Tables 1 and 2 summarize the used data and the primary sources. The main secondary sources were *Parks (1982)*, *Ogle (2017)* and the supporting information of *West, Brown & Enquist (2001)*. The authors supplemented them by data from other literature sources and from personal communications. Data in diagrams were retrieved by means of digitalization (Digitize-It of Bormisoft®). Obvious outliers were removed; this applied to data set #10 (Bull Trout).

Only data sets with $N = 6$ or more points of time were considered. Amongst data sets removed for this reason were fish data about Channel Darter from *Reid (2004)* and Creek Chub from *Quist, Pegg & De Vries (2012)*. In average, data sets had 23 points of time (maximum 110).

With respect to data collection, the paper distinguishes natural and controlled data. Most fish data were natural data. For non-fish only controlled data were collected and natural data were removed (Bottlenose Dolphin from *Read et al., 1993*; Burchell's Zebra from *Smuts, 1975*; Muntjak Deer from *Pei, 1996*). Thereby, controlled data were based on repeated measurements of the same animals (pets, aquarium fish, farmed animals, laboratory animals). The age of the animals was known, the food intake was controlled and they could easily be grouped by objective factors (e.g., sex, strain). Natural data came from hunting, fishing or capturing of wildlife, whereby in general animal age was estimated (e.g., otolith analysis). Thereby, e.g., spawning time caused additional age uncertainties for fish (*Datta & Blanchard, 2016*).

The tables inform also about the number of data points for each data set, counting the points of time. At each point of time the weights from one to several thousand animals (e.g., data set #50) were averaged. Specifically, data sets #47 and 55 were growth data from repeated measurements of a single animal. All other controlled data were average values of repeated measurements of groups of animals (group size in general 15–30 animals.) The natural data averaged over non-repeated measurements of different animals.

The authors considered only age-mass or age-length data; the latter only for fish. For most fish (see Table 1) sources provided (average) lengths of different fish of approximately the same age. In order to use data of the same format, length data were transformed into mean-weight-at-time data. Empirical evidence for fish suggested that mass may be related

**Table 1  Numbering and sources of the fish data.**

| No | Name | | Data | Comment | Source |
|---|---|---|---|---|---|
| 1 | Anchoveta | *Engraulis ringens* | 46 | | *Cubillos et al. (2001)* |
| 2 | Araucanian Herring | *Strangomera bentincki* | 36 | N, TL | |
| 3 | Atlantic (Arctic) Cod | *Gadus morhua* | 19 | | *Jørgensen (1992)* |
| 4 | Black Drum | *Pogonias cromis* | 8 | N, TL, F | *Ogle (2017)* |
| 5 | Blue Catfish | *Ictalurus furcatus* | 21 | N, TL | *Maceina (2007)* |
| 6 | Australian Bonito | *Sarda australis* | 33 | N, TL, F | *Stewart et al. (2013)* |
| 7 | | | 20 | N, TL, M | |
| 8 | Sea Trout & Rainbow Trout | *Salmo trutta & Oncorhynchus mykiss* | 9 | | *Ogle (2017)* |
| 9 | Sea (Brown) Trout | *Salmo trutta fario* | 15 | | *Abad (1982)* |
| 10 | Bull Trout | *Salvelinus confluentis* | 14[*] | | *Parker et al. (2007)* |
| 11 | Cabezon | *Scorpaenichthys marmoratus* | 13 | N, TL | *Ogle (2017)* |
| 12 | Atlantic Croaker | *Micropogonias undulatus* | 10 | | |
| 13 | European Perch | *Perca fluviatilis* | 8 | | *Mooij, Van Rooij & Wijnhoven (1999)* |
| 14 | Guppy | *Poecilia reticulate* | 14 | C, TW | *Brown & Rothery (1993)* |
| 15 | Jackass Morwang | *Nemadactylus macropterus* | 16 | | *Ogle (2017)* |
| 16 | Jonubi | *Chalcalburnus mossulensis* | 6 | N, TL | *Yildirim et al. (2003)* |
| 17 | Lake Erie Walleye | *Sander vitreus* | 20 | | |
| 18 | Arctic Lake Trout | *Salvelinus namaycush* | 17 | N, TL, F | *Ogle (2017)* |
| 19 | | | 19 | N, TL, M | |
| 20 | Longjaw Cisco | *Coregonus alpenae* (extinct) | 8 | | *Jobes (1946)* |
| 21 | Red Drum | *Sciaenops ocellatus* | 42 | | *Vaughan & Helser (1990)* |
| 22 | Redbreast Tilapia | *Coptodon rendalli* | 19 | N, TL | *Moreau (1979)* |
| 23 | Rock Bass | *Ambloplites rupestris* | 9 | | *Wolfert (1980)* |
| 24 | Round Whitefish | *Prosopium cylindraceum* | 9 | | *Bailey (1963)* |
| 25 | Sockeye Salmon | *Oncorhynchus nerka* | 24 | N, TW | *West, Brown & Enquist (2001)* |
| 26 | Sardine | *Strangomera bentincki* | 37 | N, TL | *Cubillos et al. (2001)* |
| 27 | Siscowet Lake Trout | *Salvelinus namaycush* | 15 | N, TL, F | *Ogle (2017)* |
| 28 | | | 10 | N, TL, M | |
| 29 | Spotted Sucker | *Minytrema melanops* | 7 | | *Grabowski et al. (2012)* |
| 30 | Atlantic Bluefin Tuna | *Thunnus thynnus* | 14 | | *Krüger (1973)* |
| 31 | Troutperch | *Percopsis omsicomaycus* | 8 | N, TL | *House & Wells (1973)* |
| 32 | Virgina Spot | *Leiostomus xanthurus* | 6 | | *Ogle (2017)* |
| 33 | Walleye Pollock | *Theragra chalcogramma* | 15 | | *Ianelli et al. (2011)* |
| 34 | White Grunt | *Haemulon plumierii* | 110 | N, TL, F | *Araujo & Martins (2007)* |
| 35 | | | 104 | N, TL, M | |
| 36 | Zebrafish | *Danio rerio* | 7 | C, TW, M | *Gomez-Requeni et al. (2010)* |
| 37 | | | 7 | C, TL, L | *Kaushik, Georga & Koumoundouros (2011)* |

**Notes.**
Data: number of time points; *, time points minus outlier (removed by the authors); comment: C/N, controlled/natural data; TL/TW, time length/weight data, F/M/L, female/-male/larvae.
**Table 2** Numbering and sources of non-fish time-weight data from controlled experiments.

| No | Name | | Data | Comment | Source |
|----|------|--|------|---------|--------|
| 38 | Cattle | *Bos primigenius Taurus* | 21 | Strain unknown | *Brody (1945)* |
| 39 | | | 42 | Friesian | *Parks (1982)* |
| 40 | | | 19 | Strain unknown | *Brody (1945)* |
| 41 | | | 30 | Apollo | *Parks (1982)* |
| 42 | Chicken | *Gallus gallus domesticus* | 11 | Rhode Island | *Grossman (1969)* |
| 43 | | | 30 | Ross Fryer | *Parks (1982)* |
| 44 | | | 20 | X33 strain | |
| 45 | | | 20 | X38 strain | *Ricard (1975)* |
| 46 | | | 20 | X44 strain | |
| 47 | Dog | *Canis lupus familiaris* | 23 | Rhodesian Ridgeback | (E Schläger, pers. comm., 2016) |
| 48 | | | 60 | Great Dane, F | *Parks (1982)* |
| 49 | | | 60 | Great Dane, M | |
| 50 | Domestic Pig | *Sus scrofa domestica* | 100 | Various strains | *Renner-Martin et al. (2016)* |
| 51 | Cricket | *Acheta domesticus* | 11 | | |
| 52 | | *Gryllus assimilis* | 11 | Larvae | *Sturm (2003)* |
| 53 | | *Teleogryllus commodus* | 11 | | |
| 54 | Heron | *Ardea cinerea* | 12 | | *Owen (1960)* |
| 55 | Ball Python | *Python regius* | 14 | | (F Bader, pers. comm., 2016) |
| 56 | Rat | *Rattus rattus* | 21 | Albino | *Parks (1982)* |
| 57 | | | 17 | Strain unknown | *Brody (1945)* |
| 58 | Robin | *Erithacus rubecula* | 12 | | *Owen (1960)* |
| 59 | Shrew | *Sorex cinereus* | 12 | | *Forsyth (1976)* |
| 60 | Shrimp | *Mysis mixta* | 7 | | *Rudstam (1989)* |

**Notes.**
Data: number of time points; comment: F/M, female/male, if for the same species data sets of male and female animals were considered (all data sets except #60 were unisex).

to length by an allometric power relation $m(t) = c \cdot l(t)^p$ with $2.5 < p < 3.5$ and some constant $c$ (*Pauly, 1979*; *Anderson & Neumann, 1996*). There were slight differences in the parameters $c$ and $p$, depending on which concept of length was used (*Holden & Raitt, 1974*: standard length, fork length, total length). For simplicity, the paper approximated mass by the third power of length. This convention was in line with *Von Bertalanffy (1934)*, *Von Bertalanffy (1957)* and it avoided mixing up information from different sources about time-length and length-mass relations.

The search for data aimed at capturing the full growth phase, from an early point in life (birth) till the end of growth (e.g., sexual maturation). For otherwise (Fig. 2), the modeling of a growth curve would depend on extrapolation. In particular, as *Knight (1968)* suggested, if $m_{max}$ was excessive compared to the maximal observed weight $m_{obs}$, then the model curve would not be supported by the data. Therefore, this paper considered $m_{max}$ as excessive, if $m_{max} \geq 1.5 \cdot m_{obs}$, and for sigmoid model functions, if $m_{infl} \geq m_{obs}$ (Fig. 2B). The latter condition indicates that the weight $m_{infl}$ at the inflection point exceeds the maximal observed weight, whence the inflection point of the best fitting model curve is not visible in the data. (For good approximations at least three observations should be located before and after the inflection point of the model curve, respectively.). However,
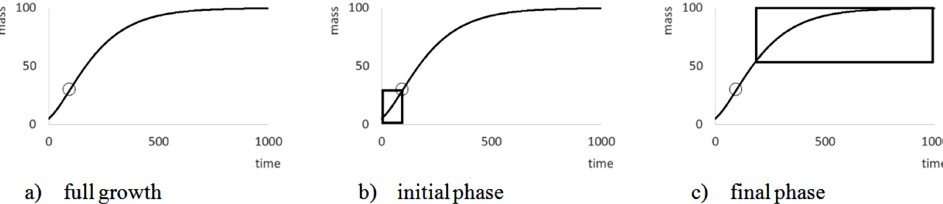

a)  full growth          b)  initial phase          c)  final phase

**Figure 2  Explanation of the meaning of 'full growth phase'.** The figure illustrates (A) the general form of VBGF ($a = 2/3$, $m_0 = 5$, $m_{max} = 100$, $q = 0.01$) showing a characteristic S-shape (circle = inflection point: $m_{infl} = 29.6$ at age $= 95.9$) over its whole range; (B) missing end-data, which suggest unbounded growth; and (C) missing data at the beginning, which suggests exponential bounded growth (exponent $a = 0$).

the authors did not discard of such data. For first, ideally a good growth model should predict the mature body mass at a juvenile age, already, and second, $m_{max}$ did not relate alone to the data, but to data fitting. The paper therefore retained such data and marked those with an excessive $m_{max}$. An exception were the Freshwater Drum data from *Bur (1984)*, removed due to an infinite asymptotic weight.

## Statistical methods and data fitting

Generally, computations were done in Microsoft® EXCEL. Optimization used its SOLVER Add-In, which implemented an iterative optimization method (Newton's method). *Casella & Berger (2001)* was used as a standard reference for statistics and XL-Stat of Addinsoft® was used for statistical computations.

Nonparametric Mann–Whitney test was used for the comparison of location parameters. (For example, mean values of optimal exponents were compared for different groups of data sets.) The Anderson-Darling test was used to assess the good fit of a normal distribution. The Fisher exact test was used for testing the significance of contingencies and the $t$-test was used to test the significance of correlations. These computations used XL-Stat.

For confidence intervals, Clopper–Pearson confidence limits were used, as these are conservative (higher confidence than stated) and suitable for small samples. Given a sample of size $M$ (the number of data sets) and amongst them $m$ ones with a specific property (the number of data sets not rejecting a certain exponent), then using the beta distribution and EXCEL notation, the one-sided lower 90%-confidence limit and the upper 90%-confidence limit for the frequency of this property in the population were $1 - \text{BETA.INV}(0.9; M - m + 1; m)$ and $\text{BETA.INV}(0.9; m + 1; M - m)$. Excel notation is used as there is no standard mathematical notation for the involved functions.

For data fitting, the paper used nonlinear regression by means of the method of least squares. As Fig. 3 illustrates, for certain data sets it was meaningful to identify optimal exponents by this method, as the resulting fit to the data of the model curves (1) was evidently different for different exponents. (Thereby, except for the exponent, all other model parameters were selected to ensure a best fit.) This approach assumed age was controlled and weight observations came from a random sample of animals with a given
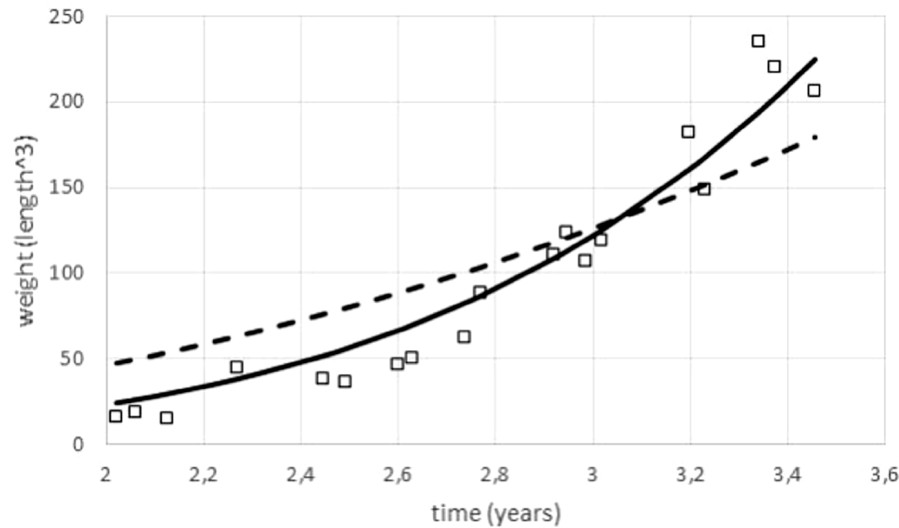

**Figure 3** **Comparing the fit of model (1) with different exponents to growth data.** Figure generated in Microsoft EXCEL, based on data set #3 and the least squares fit to these data of model (1); squares: data (weight as length³); dashed line: model curve for the exponent $a = 0.67$ ($m_0 = 0.14$, $m_{max} = 23{,}538$, $q = 0.18$); thick line: model curve for the optimal exponent $a = 0.99$ ($m_0 = 0.14$, $m_{max} = 23{,}538$, $q = 26.4$).

age. Implicitly, the method of least squares is a maximum likelihood estimate assuming normally distributed deviations of the data from the model curve (residuals).

## Alternative regression models

As there is a large body of nonlinear regression literature, a few explanations are added to justify the above choice of methods. For instance, in fisheries literature it has been recommended to assume that weight was controlled and age observations came from a random sample of animals with a given weight (*Sparre & Venema, 1988*; *Piner, Hui-Hua & Maunder, 2016*). For the paper, this approach was rejected, as for the present data, all non-fish data were controlled for age and for most non-fish data there was no control. However, as explained below, the alternative approach was used for the numerical computations to obtain a first approximation of the best fit parameters.

There are alternative heteroscedastic growth models assuming a larger variance for higher weights. (For example, for lognormally distributed weights the standard deviation is proportional to the expected weight.) An increase in variance may also be caused by the availability of fewer data for older animals. (This relates to the dependency of the variance of the mean value on sample size.) However, owing to relatively small sample sizes (counting the number of different points of time) distribution fit tests did not refute the assumption of normally distributed residuals for the present data sets. Therefore, the paper used the simpler least squares method.

Literature in the context of feeding experiments applies also improvements of regression models, such as mixed-effect models (e.g., *Strathe et al., 2010*). However, as the purpose of such models would be the identification of explanatory factors for growth (e.g., gender), such models require highly controlled experiments, distinguishing the relevant factors in

the data. For most of the present data such information was not available. (For instance, many data sets did average over male and female fish.) Therefore, the present paper did not study such more complex models.

In a different direction the model fit could be simplified by using literature values for certain parameters. For instance, for fishes the mass at age 0 ($m_0$) is largely similar (*Andersen, 1969*), whence a literature value for $m_0$ could be used instead of optimizing this parameter. However, by this approach the weight at $t = 0$ would be exceptional in comparison to other data points, which it is not, whence also in fishery literature the parameter $m_0$ is usually optimized. Further, for non-fish species the neonate weight is more variable and the authors did not wish to treat data fitting differently for different species.

## Numerical approach towards data fitting

For most data sets a straightforward approach towards data fitting failed due to numerical instability. (Data fitting means searching for values $a$, $m_0$, $m_{max}$, and $q$, where the sum of squared residuals is minimal.) The authors therefore did optimization in two steps:

In the first step, given an exponent $0 < a < 1$, optimal parameter values $m_0$, $m_{max}$, and $q > 0$ for model (2) were sought to minimize the sum of squared residuals between the data points and the model function; the squared residual for the $n$th data point ($t_n$, $m_n$) is $(m_n - m(t_n))^2$ and $SSR = SSR(a)$ is the sum of these residuals.

In the second step, the paper sought to obtain an optimal exponent, where $SSR = SSR(a)$ was minimal; the desired accuracy for the exponent was 0.01. Therefore, using a macro the optimization was repeated for each exponent $a = 0, 0.01, 0.02, \ldots$, and 0.99, resulting in 100 models for each dataset. The resulting minimal values of $SSR(a)$ were tabulated. Summarizing, this defined an optimal exponent $a_{opt} < 1$ and optimal parameter values $m_0$, $m_{max}$, and $q$ for this exponent.

In view of numerical instability, the convergence of the first step of optimization (using the SOLVER) required good initial estimates of the optimum parameters. These were computed by adapting a graphical method (Fig. 4), the Bertalanffy-Beverton plot (*Von Bertalanffy, 1934*). It was based on two ideas: fitting the inverse of function (2) to the weight-time data (rather than fitting to time-weight data) and finding a transformation of the weight-time data which allows to use a linear regression for this exercise.

Summarizing, this method aims at computing an optimal fit of the weight-time data to the inverse function of (2). It is described by Eq. (3) for $t = t(m)$; ln is the natural logarithm function:

$$t = \frac{f(m_0) - f(m)}{q} \quad \text{where } f(x) = \frac{\ln(1 - (x/m_{max})^{1-a})}{1-a} \quad \text{for } x < m_{max}. \tag{3}$$

Collecting terms not depending on $m$ simplified this to Eq. (4):

$$t = A + B \cdot f(m) \quad \text{with } A = f(m_0)/q, B = -1/q. \tag{4}$$

The Bertalanffy-Beverton plot (Fig. 4) was motivated by Eq. (4): plotting, for all data points, time $t$ over transformed mass $f(m)$ should result in approximately a straight line. Therefore, $A$, $B$ in formula Eq. (4) are the coefficients of a linear regression line $t = A + B \cdot u$

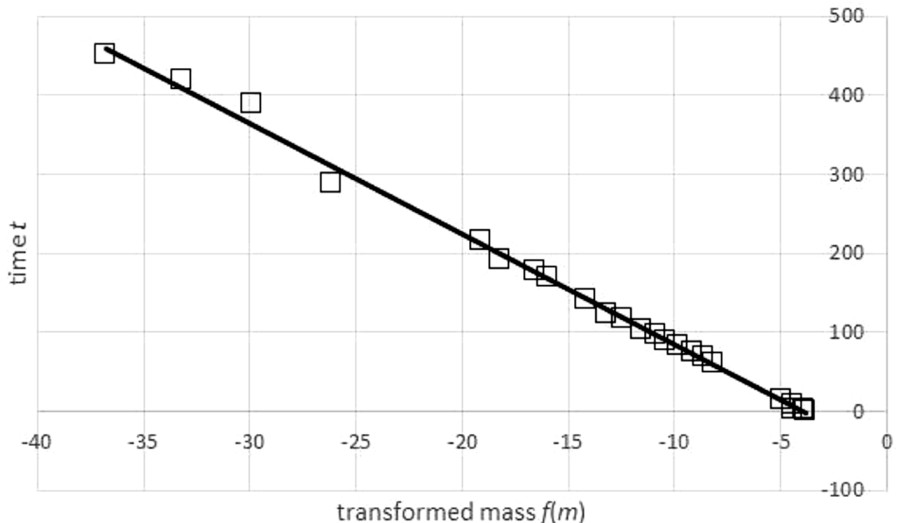

**Figure 4  Transformation of time-mass-data and a regression line for the transformed data set.** Generalized Bertalanffy-Beverton plot generated in Microsoft EXCEL, based on data set #47; squares: transformation of data $(t, m)$ into $(u, t) = (f(m), t)$; line: regression line $t = A + B \cdot u$ with $A = -55.64$ and $B = -14.03$. The function $f$ was defined in Eq. (3) using the exponent $a = 0.83$ and assuming an asymptotic weight limit $m_{max} = 32,163$ g. The transformation required $m_{max}$ to exceed the maximal observed weight (31.8 kg), as otherwise the transformation would not be defined for all data points.

that is fitted to transformed data $(u_n, t_n) = (f(m_n), t_n)$, using the function $f$ of Eq. (3). In EXCEL, $A$ and $B$ were computed with the LINEST function.

However, by Eq. (3), this transformation depends on assuming, in addition to the given exponent $a$, a value for the (true) asymptotic weight limit $m_{max}$. As $m_{max}$ was not known, the goodness of fit of the regression line was evaluated for different values of $m_{max}$, using the sum of squared residuals $SSR_{inv}$ $(m_{max})$ for the transformed data. This defined a function in $m_{max}$. (In EXCEL, it was again computed by the LINEST function.) It was assumed that this sum $SSR_{inv}$ $(m_{max})$ was minimal for the true $m_{max}$.

The search for $m_{max}$ with a minimal sum $SSR_{inv}$ $(m_{max})$ caused no numeric problems: As illustrated by Fig. 5, for all data the function $SSR_{inv}$ $(m_{max})$ decreased rapidly for $m_{max}$ near the maximum $m_{obs}$ of the observed data (observed maximum) and it was flat for larger values of $m_{max}$. The SOLVER Add-In minimized this function iteratively (using as a starting value $1.01 \cdot m_{obs}$) under the additional constraints $m_{max} > m_{obs}$, as otherwise $f(m)$ would not be defined for the larger data, and $m_{max} < 100 \cdot m_{obs}$, as in view of the flat curve larger values of $m_{max}$ did not substantially reduce $SSR_{inv}$. (For the purpose of optimization it did not matter that an excessive $m_{max}$ was not always empirically meaningful.) As the optimization used exact formulae for $A$, $B$ (LINEST function) and as it was done in one dimension (seeking $m_{max}$ with minimal $SSR_{inv}$), it could be performed fast and with high precision.

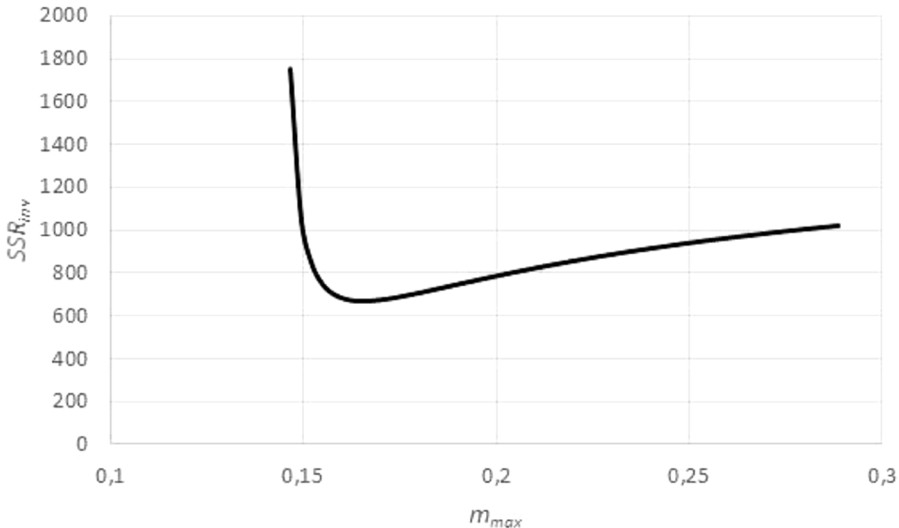

**Figure 5 Optimizing the asymptotic weight limit (fit to weight-time data).** Figure generated in Microsoft EXCEL, based on data set #14, plotting the sum of squared residuals $SSR_{inv}$ (fit to the weight-time data) in dependency on $m_{max}$ for an exponent $a = 0.67$. The minimum was attained for $m_{max} = 0.165$ g (maximal observed weight: $m_{obs} = 0.145$ g) resulting in the estimates $q = 0.1$/day and $m_0 = 0.03$ g. These were used as a starting value for the minimization of $SSR$ (fit to the time-weight data). The resulting optimal parameters for $a = 0.67$ were $q = 0.139$/day, $m_0 = 0.002$ g and $m_{max} = 0.149$ g.

Using this preliminary optimization, the following initial estimates for the fit of model (2) to the data were used: the given exponent $a$, the above optimized $m_{max}$, $q = -1/B$, and for $m_0$ the least observed weight $m_{min}$.

These values $m_0$, $m_{max}$, and $q$ were used as starting values for the first optimization step, the iterative optimization of $SSR = SSR$ $(a)$, given the exponent $a$. For all data sets, the SOLVER converged. In order to hinder parameter values from moving from the reasonable starting values into a region of divergence, the above constraints $m_{obs} < m_{max} < 100 \cdot m_{obs}$ were retained and $m_0 > m_{min}/100$ was added.

As for a computationally simpler method of obtaining initial estimates for the best fit parameters, there is a large body of literature using the Walford plot for data fitting (e.g., *Espino-Barr et al., 2015*), which was explained in Fig. 1. However, that method did not always provide good initial estimates.

## Model comparison

In order to compare the goodness of fit, for each data set the 100 models corresponding to different exponents $a < 1$ were assessed by means of the Akaike information criterion (*Akaike, 1974*; *Burnham & Anderson, 2002*; *Motulsky & Christopoulos, 2003*), using an index $AIC_c$ for small sample sizes. It was computed from $SSR(a) =$ the sum of squared residuals, $N =$ number of data points, and $K = 4 =$ number of optimized parameters (namely $m_0$, $m_{max}$, $q$ and implicitly $SSR$). The number of data points essentially counted, for how many points of time (ages) there were data. (If there were several observations for the same age, as e.g., for reported average values for groups of animals, then this was

counted as one data point.)

$$AIC(a) = N \ln\left(\frac{SSR(a)}{N}\right) + 2 \cdot K + \frac{K \cdot (K+1)}{N-K-1} \qquad (5)$$

$$prob(a) = \frac{e^{-\Delta/2}}{1 + e^{-\Delta/2}}, \text{ where } \Delta = AIC(a) - AIC(a_{\mathrm{opt}}) > 0. \qquad (6)$$

Formula (6) gives the probability $prob$ (Akaike weight) that the model with exponent $a$ was true, when compared with the better fitting model with exponent $a_{\mathrm{opt}}$, assuming that either $a$ or $a_{\mathrm{opt}}$ would be the true exponent. Assuming that one of the two exponents $a$ or $a_{\mathrm{opt}}$ is true, then $prob(a) + prob(a_{\mathrm{opt}}) = 100\%$. If the better fitting model is more likely to be true, $prob(a) \le prob(a_{\mathrm{opt}})$. It follows that $prob(a) \le 50\%$ and $prob(a_{\mathrm{opt}}) = 50\%$.

For the present paper, an exponent $a$ is refuted (for a data set), if in comparison to $a_{\mathrm{opt}}$ its Akaike weight is below 2.5%. (Thus, 'below 2.5%' defines the 5% of the index values with the worst fit.) The paper uses the Akaike weight for this definition, as other measures for the goodness of fit (e.g., $SSR$) make sense only with respect to one given data set, while the present paper seeks a definition applicable across different data sets. Further, this notion of refutation uses the Akaike weight merely as an index for the goodness of fit, whence a notion of truth is not required. In particular, the paper makes no assumption, whether one of the exponents $a$ or $a_{\mathrm{opt}}$ is true, as this is not needed for a refutation owing to a poor fit: if a model (defined from an exponent $a$) is refuted, as it fares poorly amongst its 'peers', it is sensible to refute it also for any larger group of models. Further, if model (1) were false, then the refutation of such an exponent $a$ would be sensible anyway.

The Akaike weight was used to quantify the variability of the exponent, figuring out over which range of exponents the best fit to the data did not change significantly. However, as this weight was sensitive to outliers, in a preparatory step outliers had to be removed; see Fig. 6. The curve $prob(a)$ had its peak at $a_{\mathrm{opt}}$ and it was increasing/decreasing for lower/higher values of $a$. Therefore, the non-refuted exponents formed an interval.

FNR is the fraction of non-refuted exponents amongst the 100 considered exponents ($a = 0, 0.1, \ldots, 0.99$). It measures, how strong the optimal exponent is determined by the data; the lower the FNR the less uncertainty remains about the value of the exponent $a$. FNR was computed as the difference between High and Low of Table 3 plus 0.01 (as $a_{\mathrm{opt}}$ is not refuted), whereby High and Low were the highest and lowest non-refuted exponents.

## RESULTS

Table 3 summarizes the fit of model (1) to 60 data sets. There occurred three issues.

First, there was a high variability in the exponent $a$, which for 18 data sets could take any value (between 0 and 0.99) without affecting the fit to the data significantly (when the other model parameters were optimized); FNR = 1 for these data sets. Consequently, it could not be assured that a biologically correct exponent could be identified from data fitting.

Second, there were problems with unrealistic parameter values, specifically for the estimated mature body mass $m_{\mathrm{max}}$. For 13 data sets, it exceeded the maximally observed weight by 50% ($m_{\mathrm{max}}/m_{\mathrm{obs}} \ge 1.5$), whence boundedness was not evident from the data,
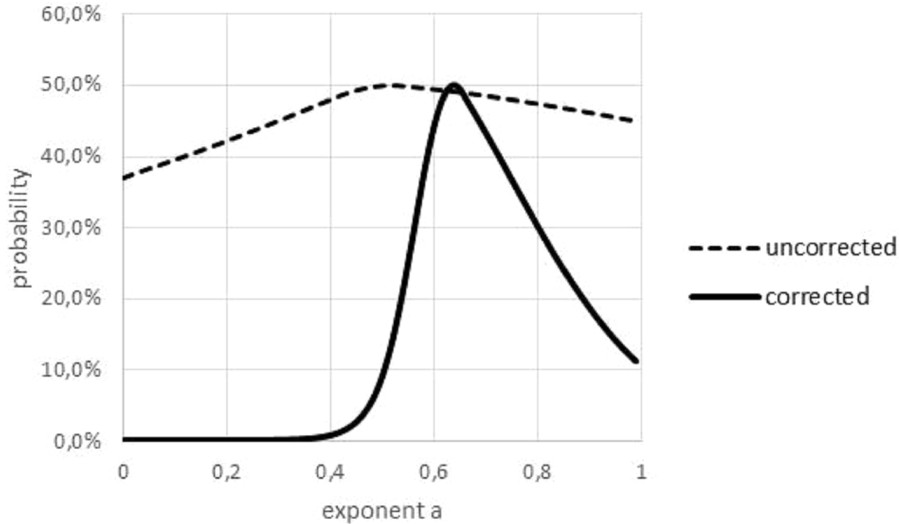

**Figure 6 Akaike weights for different exponents, for data with and without outliers.** Graphical multi-model comparison, generated in Microsoft EXCEL, based on data set #10 with and without outlier. The Akaike weight *prob(a)* for model (1) with exponent *a* was computed in comparison with the optimal exponent; dashed/fat line Akaike weights for data with/without outlier. Without outlier, the optimal exponent became larger (0.64) and exponents with poor fit could be discerned more easily.

and for 7 data sets with sigmoid model curves ($a_{opt} > 0$) the weight at the inflection point exceeded the maximally observed weight ($m_{infl} \geq m_{obs}$), whence the sigmoid shape of the model curve was not discernible from the data. For one data set (#50, domestic pig), the excessive $m_{max}$ could be explained in part from data collection, as it ended long before sexual maturity had been reached.

Third, there was a concentration of the optimal parameters on the boundary of the parameter space. (If maximum likelihood parameters assume values on the boundary of the parameter space, then the normal distribution of the optimal parameters is no longer assured). For 15 data sets $m_{max} = m_{obs}$ assumed the lower bound for optimization. Moreover, for 24 data sets, the exponent was optimal for the largest value $a = 0.99$; for 5 data sets, the exponent was optimal for the lowest value $a = 0$ (not sigmoid). The tendency towards higher exponents persisted, if the variability was taken into account (Fig. 7): for 55 data sets the interval of non-refutation reached the upper bound 'High' $= 0.99$. Further, for any $a \leq 0.01$, at least 50% of data sets would reject this low exponent, while for any $a \geq 0.89$, at least 80% of data sets would not reject this high exponent. (For each statement 95% confidence was confirmed, assuming data sets selected at random by the same strategy, as for this paper.)

Summarizing, model (1) may be suitable to describe the growth of certain species, but for 49 (82%) of the 60 data sets (95% confidence: at least 72% of data sets) there were issues for optimization with unrealistic model parameters, with parameters on the boundary of the parameter space, or with elusive parameters that due to a high variability could not be determined accurately from the data. This indicated that for a substantial fraction of

**Table 3  Optimal exponents and range of acceptable exponents.**

| No | $a_{opt}$ | max | Non-refutation Low | High | No | $a_{opt}$ | max | Non-refutation Low | High | No | $a_{opt}$ | max | Non-refutation Low | High |
|----|-----------|-----|-----|------|----|-----------|-----|-----|------|----|-----------|-----|-----|------|
| 01 | 0.83 | NO | 0.43 | 0.99 | 21 | 0 | NO | 0 | 0.99 | 41 | 0.85 | C | 0.65 | 0.99 |
| 02 | 0.76 |    | 0.11 | 0.99 | 22 | 0.62 |    | 0.51 | 0.99 | 42 | 0.99 |    | 0.75 | 0.99 |
| 03 | 0.99 | A  | 0.9  | 0.99 | 23 | 0.57 | B  | 0    | 0.99 | 43 | 0.99 |    | 0.79 | 0.99 |
| 04 | 0.39 | NO | 0.21 | 0.99 | 24 | 0.51 |    | 0.35 | 0.99 | 44 | 0.88 | NO | 0.78 | 0.99 |
| 05 | 0.75 | C  | 0    | 0.99 | 25 | 0.91 | NO | 0.64 | 0.99 | 45 | 0.99 |    | 0.9  | 0.99 |
| 06 | 0    |    | 0    | 0.99 | 26 | 0.99 |    | 0.34 | 0.99 | 46 | 0.99 |    | 0.85 | 0.99 |
| 07 | 0    | NO | 0    | 0.99 | 27 | 0.91 | C  | 0.01 | 0.99 | 47 | 0.83 |    | 0.68 | 0.99 |
| 08 | 0.99 |    | 0    | 0.99 | 28 | 0.99 | NO | 0.02 | 0.99 | 48 | 0.84 | C  | 0.58 | 0.99 |
| 09 | 0.66 | A  | 0.54 | 0.99 | 29 | 0.99 | A  | 0    | 0.99 | 49 | 0.99 |    | 0.68 | 0.99 |
| 10 | 0.64 | NO | 0.44 | 0.99 | 30 | 0.11 | B  | 0    | 0.99 | 50 | 0.45 | A  | 0.37 | 0.93 |
| 11 | 0.6  |    | 0    | 0.99 | 31 | 0.19 | A  | 0    | 0.99 | 51 | 0.99 |    | 0.8  | 0.99 |
| 12 | 0    | B  | 0    | 0.99 | 32 | 0.99 | NO | 0    | 0.99 | 52 | 0.94 | NO | 0.75 | 0.99 |
| 13 | 0.8  | C  | 0    | 0.99 | 33 | 0.37 |    | 0.22 | 0.67 | 53 | 0.99 |    | 0.89 | 0.99 |
| 14 | 0.99 | NO | 0.51 | 0.99 | 34 | 0.01 | C  | 0.01 | 0.99 | 54 | 0.99 |    | 0.78 | 0.99 |
| 15 | 0.99 | A  | 0    | 0.99 | 35 | 0    |    | 0    | 0.99 | 55 | 0.99 | C  | 0.57 | 0.99 |
| 16 | 0.99 | B  | 0.08 | 0.99 | 36 | 0.99 | NO | 0.66 | 0.99 | 56 | 0.14 | NO | 0    | 0.52 |
| 17 | 0.21 |    | 0    | 0.99 | 37 | 0.99 |    | 0.57 | 0.99 | 57 | 0.78 | C  | 0.55 | 0.99 |
| 18 | 0.69 | C  | 0.11 | 0.99 | 38 | 0.24 | C  | 0    | 0.54 | 58 | 0.99 |    | 0.16 | 0.99 |
| 19 | 0.28 |    | 0    | 0.99 | 39 | 0.2  | B  | 0.02 | 0.38 | 59 | 0.99 | NO | 0.3  | 0.99 |
| 20 | 0.99 | A  | 0.38 | 0.99 | 40 | 0.17 | C  | 0    | 0.99 | 60 | 0.99 |    | 0.76 | 0.99 |

**Notes.**

Data sets numbered as in Tables 1 and 2; abbreviations for max: 'NO' no problem with $m_{max}$, 'A' and 'B' excessive $m_{max}$, because $m_{infl} \geq m_{obs}$, and because $m_{max} \geq 1.5$. $m_{obs}$, respectively, and 'C' $m_{max} = m_{obs}$; $a_{opt}$ = exponent with the best fit of model (1) to the data; non-refutation: lower and upper bounds of the interval consisting of those of exponents that in comparison to $a_{opt}$ were not to be refuted (Akaike weight 2.5% or higher).

species a more general or a completely different class of models might be needed to describe growth more realistically.

In particular, the data did not support the claims from literature about the existence of a universal exponent for model (1) that would suffice to describe the growth of a large taxonomic group of animals. On the contrary, also for the 11 data sets not showing the above issues (#1, 2, 4, 10, 22, 25, 33, 44, 47, 52, 56), the optimal exponent $a_{opt}$ varied widely and its mean value did not differ significantly between these 11 data sets and the 49 problematic ones (Mann–Whitney test, $p$-value 20.3%). Further, the data did not support the hypothesis that for each species there would be optimal exponents to describe its growth by model (1). For, the best fitting exponents could differ widely for female and male animals of the same species, as demonstrated for data sets #18–19 (female and male Lake Trout) with optimal exponents 0.69 and 0.28, respectively. Further, where e.g., fish of the same species came from different locations (different water temperatures for the Lake Trout data #18–19 and #27–28), a different pattern of growth was expected for biological reasons.

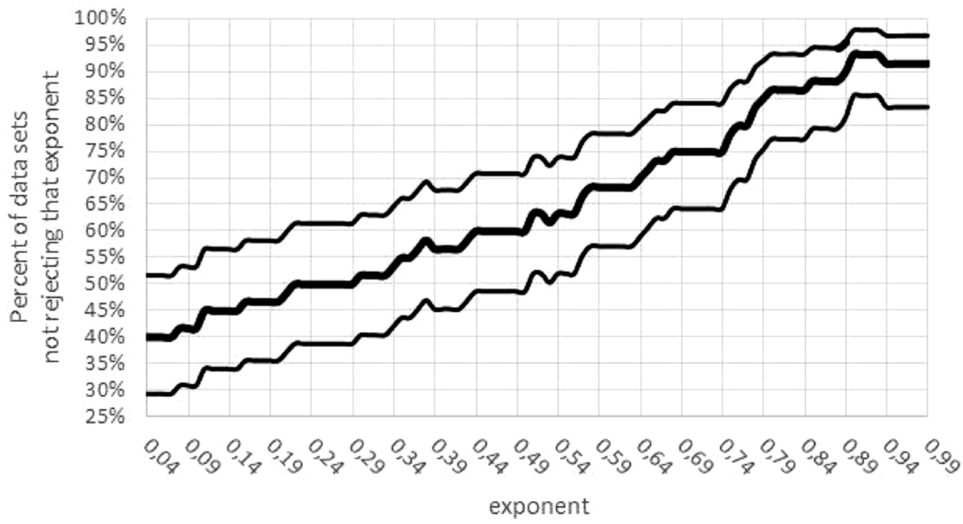

**Figure 7 Confidence intervals for the percentage of data sets not rejecting an exponent.** Figure generated in Microsoft EXCEL. Fat curve counts the percentage of how many of the 60 data sets did not reject the exponent $a$; thin curves one-sided Clopper–Pearson confidence limits (95% significance).

## DISCUSSION: EXPLAINING VARIABILITY

The observed issues, in particular the high variability of the exponent, may indicate that data fitting optimized too many parameters. However, this paper proposes an alternative explanation, namely differences in data quality. For, data for non-fish in general came from highly controlled experiments, while data for fish in general were natural data.

This claim is supported by the highly significant dependency of variability on whether the data were for fish or not: 17 of the 18 species with FNR = 1 were fish. Further, with 99.99% significance (Mann–Whitney test: $p$-value below 0.01%) the average FNR for fish (0.8) was stochastically larger than the average FNR for non-fish (0.38), even though there were also fish with small FNR (e.g., FNR = 0.1 for Artic Cod #3).

Ignoring this variability could generate artefacts. For instance, with 95% significance (Mann–Whitney test, $p$-value 4.7%) the mean value of the optimal exponents for fish (0.61) was stochastically lower than the mean value for non-fish (0.79). However, the reason for this could be variability, as with 99.99% significance the optimal exponents were negatively correlated with FNR ($t$-test: $p$-value below 0.01%); i.e., lower optimal exponents for fish were related to a higher FNR (because 'Low' became smaller). By contrast, there were no 95% significant differences between fish and non-fish for the fractions $m/m_{obs}$ and $m_{infl}/m_{obs}$.

The following discussion aims at discerning characteristic features of natural data, which did/did not affect variability.

An obvious difference between fish and non-fish data was the transformation of length to weight, which was needed for most fish. This paper used a power-law transformation $m(t) = l(t)^p$ with $p = 3$ for fish. As Fig. 8 illustrates, this convention could have affected refutations, but it could not explain a systematic bias towards easier or more difficult

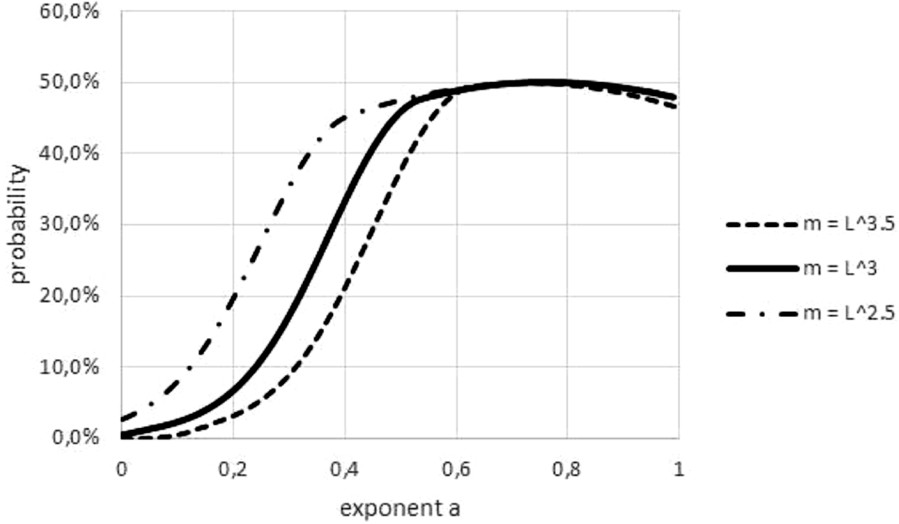

**Figure 8 Effect on the Akaike weights of using different length-mass relations.** Figure generated in Microsoft EXCEL, based data set #2, plotting Akaike weights for modifications of the data set, using different powers of length to estimate mass.

refutations. (In the figure, higher/lower values of $p$ make refutations easier/more difficult. However, for some data, the 'true' $p$ was below 3 and for others it was above 3).

Another difference between data sets was insufficient stratification for fish: for non-fish species the data for females and males of the same species were collected separately, this was not the case for all fish. In case that these groups had a different growth pattern (different optimal exponents), as for Fig. 9, the combination of data could result higher FNR. (Intuitively, combined data have higher residuals, making refutations more difficult). Similar effects could have been generated from the combination of data from different locations or from data about migrant animals, collected at the same location but not taking into account the different origin of the animals.

However, in the authors' view, the main difference seems to have been whether data were comprised of repeated measurements of the same animals (controlled data) or of single measurements of randomly chosen animals (natural data). For, the three discarded natural data sets for non-fish species showed a high variability in the exponent, as did the fish data, and the three fish data sets from controlled experiments showed a rather lower variability. (More data sets would be needed to support this view by statistical reasoning.)

The authors developed the following strategy to identify data sets, where model (1) could be expected to provide an adequate fit: data were removed, if FNR $= 1$ (maximal variability), $m_{max}/m_{obs} \geq 1.5$, $m_{infl}/m_{obs} \geq 1$ (unrealistic parameter values), $a_{opt} = 0$, $a_{opt} = 0.99$, or $m_{max}/m_{obs} = 1$ (parameters on the boundary). There remained 11 data sets. For them here was no significant contingency between fish and non-fish data. (Fisher exact test resulted in the $p$-value 100%, testing the contingency table comprised of seven selected and 30 non selected fish and of four selected and 19 non selected non-fish.) Thus, this strategy appeared to be successful in eliminating the differences between natural and

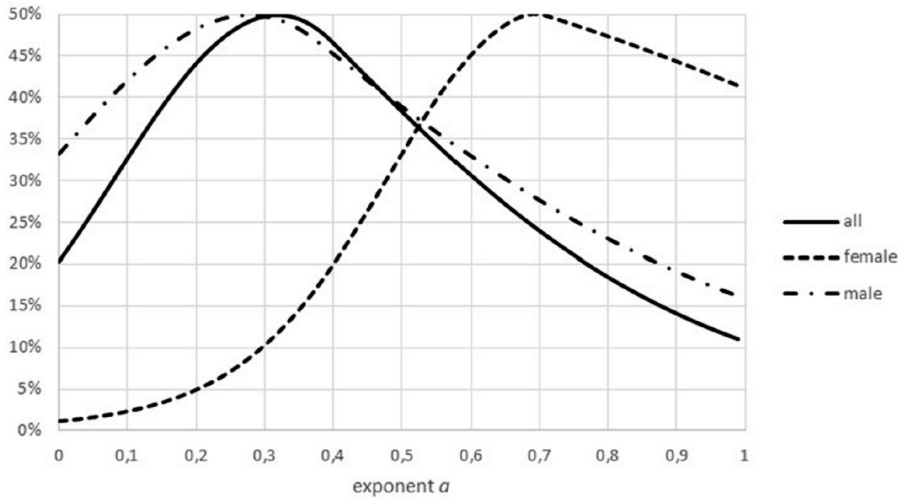

**Figure 9  Effect of combining males and females on the Akaike weights.** Figure generated in Microsoft EXCEL, based on data sets #18 and #19, plotting Akaike weights for data separated by sex and for the combined data.

controlled data, as far as data fitting was concerned. However, also for this smaller sample of data sets with apparently unproblematic data fitting there did not emerge a candidate for a universal exponent; the exponent ($a_{opt}$) took values between 0.14 and 0.94.

## CONCLUSION

The paper argued that the question, whether there exists a universal metabolic scaling exponent for the generalized Von Bertalanffy model (1) may be ill-posed, as the biological meaning of the model exponent, and the variation in this exponent, might not be clear. To this end it computed optimal exponents from a sample of 60 data sets about the growth of animals.

A closer look at data-fitting revealed serious deficiencies. There was uncertainty about the optimal exponent (variability), there were unrealistic parameter values (excessive estimates for the mature body weight) and parameters tended to assume values on the boundary of the parameter space (indication for the need of a completely different model). With respect to variability, for a large range of exponents the best fitting model curves (optimizing the other parameters) did not differ significantly in their fit to the data. (Intuitively, plots did not show noticeable deviations of the model curves with these exponents from each other). Thereby, the variability of the exponent could be related to the type of data, with natural data causing higher variability than controlled data. Thereby for fish in general only natural data were available. Ignoring these deficiencies could result in biologically unfounded statements, such as observing stochastically lower exponents for fish than for non-fish species.

However, also for data sets selected for not showing the above mentioned deficiencies, the optimal exponents computed from the data differed widely from the values proposed

in literature on the basis of biological reasoning (e.g., $a = 2/3$ or $a = 3/4$). In view of this discrepancy data fitting did not support any of the biological explanations of the exponents.

Interpreting variability positively, if the model is merely intended to summarize information from (mass) growth data by means of a few parameters, then for data sets with large variability of the exponent almost any exponent may be adequate (if it is close to 1: Fig. 7). Thus, in hindsight, also the use of VBGF to summarize fish data (e.g., FishBase) appears to be justified, unless the biological meaning of the model parameters is overstressed.

## ACKNOWLEDGEMENTS

The authors appreciate the insightful reviewers' comments that substantially improved the paper. The paper is part of the first author's doctoral thesis.

### Funding

The work of Katharina Renner-Martin was supported by a grant from the Universität für Bodenkultur. The funders had no role in study design, data collection and analysis, decision to publish, or preparation of the manuscript.

### Grant Disclosures

The following grant information was disclosed by the authors:
Universität für Bodenkultur.

### Competing Interests

The authors declare there are no competing interests.

### Author Contributions

- Katharina Renner-Martin, Norbert Brunner, Manfred Kühleitner, Werner Georg Nowak and Klaus Scheicher analyzed the data, contributed reagents/materials/analysis tools, wrote the paper, prepared figures and/or tables, reviewed drafts of the paper.

### Data Availability

The raw data is provided in the Supplemental Files.

### Supplemental Information

Supplemental information for this article can be found online at http://dx.doi.org/10.7717/peerj.4205#supplemental-information.

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
