# Peer review of "On the exponent in the Von Bertalanffy growth model"

_PeerJ, doi:10.7717/peerj.4205_

## Round 0.1 · original submission · Major Revisions

Both reviewers indicated that the study was clear and well written. One concern that requires further attention is the number of free parameters when fitting the model to the data. Having many free parameters will improve the model fit but may result in unrealistic parameter values. The authors should consider using prior knowledge to constrain the model. For example, reviewer 2 indicates that it is unrealistic to assume max to be more than twice as large as the maximum observed mass. Similar constrains may be placed on the other model parameters to ensure a better estimate of the underlying (unobserved) system parameters.

Reviewer 1 ·

Basic reporting

the English is clear and references and background context are sufficient. No issues with the structure of the papers , figures or the tables.
article met these criteria.

Experimental design

the experimental design area meets the criteria. The research question is defined and relevant. The investigation was reported and carried out to technical standards. The authors appear transparent in addressing issues with the data sets and analyses. Methods reported in sufficient detail.

Validity of the findings

the authors are clear on the validity of their findings. The authors used data from numerous sources for their analyzes and were transparent and the conclusions are well stated.
There is no over speculation.

Additional comments

Line 23 – size but also mass of the animal
Line 55 – how many of these data sets included repeated observations (measurements ) on individual animals? Really better analyses If mixed model approaches used like that of Strathe et al. ? Strathe, A. B., A. Danfær, H. Sørensen, and E. Kebreab. 2010. A multilevel nonlinear mixed-effects approach to model growth in pigs1. J. Anim. Sci. 88:638-649. doi:10.2527/jas.2009-1822

Note line 81 – BW to the 0.75 is used widely in animal sciences as metabolic size.

Line 194 -195 – did you also check enough points before and after the expected inflection point ?
Data with 1 or 2 points above or below the IP – results in greater SE for the model parameters ..
it is not clear why the transformation was done in figure 6 and how often this was done in the data sets
note the variation in Bw increases as animals grow and ordinary least squares do not account for this and using the means at each age as mentioned in line 203 --- does not really take the equal variances into account as the SE of the BW means increase as BW increases – it should be discussed what was assumed and the better methods of doing the statistics when actual data is available included mixed models and accounting for heterogeneous variances

Reviewer 2 ·

Basic reporting

The manuscript is generally well-written. Yet, I recommend authors to address the following:

1) Authors present parts of their methods and results in other sections of the manuscript. Please move lines 57-58 (including Fig 1) and lines 387-401 to the result section. Secondly, please move line 33 “this paper assumes b=1”, Lines 100-101 “Figure 2 displays ....”, lines 115-118 and lines 356-359 to the method section.

2) Multiple times the line of reasoning is unclear to me. Please clarify:
o Lines 26-30 and how this relates to the text before. Improved otolith analysis is not an application. Why do authors also write a full sentence about starfish populations? This is not related to their own work. Please broaden the first paragraph of the introduction. Potentially with text from lines 119-122.
o Explain 41-43 in more detail. Explain why authors describe the inflection point? Is this detail needed in the introduction?
o Again for lines 72-77, is this amount of detail needed in the introduction? If so, please rephrase.
o Lines 117-118. Is this comment related to the literature or to the work described in this study (if the latter, please move to method section).
o Lines 122-126 change to: In addition, a search in google scholar identified approximately 24800 papers related to the von Bertalanffy growth model in fish (using the search terms “...”). The use of VBGF in the literature was also specifically surveyed for elasmobranch species, showing that the VBGF model was studied twice as often as any other model (Smart et al. 2016).
o Lines 146-148: what is the “weight” at the inflection point. Secondly, how can it vary 0% of mmax. Please clarify.
o Lines 368-369 Why is this sentence included? For which type of animals do authors expect sex change? Has this been reported in the data they use?

3) Authors use Excel annotation to describe how they analyzed their data (e.g. line 225, 230, 232-235, 249). This makes the text unclear. Please rephrase (to standard text annotation) and include all Excel code in supplement.

4) Change lines 155-193 from text into a table with numbers, names, references and additional information on the growth data (number of time steps (days, months, years), whether the data is based on an individual growth trajectory or the mean of a group and whether the original data is in length or weight).

Experimental design

The authors did a very good job to get an extensive dataset on growth to analyze their research question. The question is well-defined and highly relevant. No further comments.

Validity of the findings

The work essentially shows that there is large variability in parameter a when other parameters (mmax, q and m0) are also optimized. This highlights that different combinations of these 4 parameters fit the data best, but that the data quality is not good enough / the growth model too flexible (with many free parameters) to quantify parameter a (and q, mmax and m0) in a consistent way. What is interesting is that in many species parameter a can vary over a very large range (between 0-0.99) without affecting the fit to the data. This is to me the main finding of the paper and it is also described by the authors in lines 410-414.

Given that combinations of the other parameters affect parameter a so much, I have some suggestions/comments about the other parameters:
o The mass at age 0 (m0) is largely similar for fishes (see Andersen et al 2008 Theoretical Population Biology), while m0 might be known for the other species in the dataset. It is perhaps better to derive m0 from the literature (instead of optimizing this parameter). This will reduce the flexibility of the optimization, since there are less “free” parameters, and hopefully lower the parameter space that can fit the data in an adequate way.
o Authors have constrained Mmax and assumed that this parameter needs to be higher than the maximum observed mass. Authors state mmax can exceed the maximum observed mass up to 100 times (lines 275-276). I’m not convinced with this as any mmax that is ~2-100 times larger than the observed maximum basically shows that the data is not strong enough to do the analysis (no data point close to mmax). This is very important as uncertainty in mmax is directly affecting parameter a (as these parameters are dependent on each other in the model). Hence, I would suggest removing all the species that have a mmax that is much higher (e.g. 2 x as high) than the maximum mass observed in the data.
o Finally, it is not clear to me why authors did not optimize parameter b. Is there evidence in the literature that b should be 1? How will variation in b affect the optimization of parameter a?

Furthermore, I suggest to focus less on the fact that a=0.67 is weakly universal for fish, also because the percentage of non-rejection (fig 8) varies weakly when parameter a is larger than 0.65 (probably the effect of 1 or 2 fishes). If authors would have chosen a threshold of 80% or 85% (see line 106), all values higher than 0.65 of parameter a would be defined as weakly universal. It now seems as if VBGF is supported by the data, while the 0.75 exponent not. However, the differences are very small and as such I find the choice of threshold (90%) a bit misleading.

Lastly, the difference between fish and non-fish is also a difference between highly controlled experiments (where food is not limiting) (non-fish) versus natural conditions (most fish). As such, I would not do a comparison as in Table 3 and only show that both fish and non-fish species have a large range of values for parameter a that fit the data adequately. The difference between fish and non-fish seems to be an artefact and authors provide no biological explanation why fish potentially differ from other taxonomic groups. As such, I also recommend removing lines 18-19 in the abstract.

Additional comments

Line 47 Can authors provide more explanation what (mt/mmax)^1-a exactly is. And what are the units (mass per time)?

Line 83 please explain “noisy data may hide” or remove.

Lines 201-202 please clarify this sentence.

Lines 212-215 please explain how this is standardized across the 60 datasets?

Lines 217 Why has parameter b a value of 2 in Figure 5? Previously it was stated that the value of parameter b is assumed to be 1, please explain?

Line 295 remove “that”

Lines 294-298 maybe add “ FNR shows the parameter space that fits the data adequately. The lower the FNR the less uncertain the value of parameter a”.

Lines 299-300 please clarify where this is justified (discussion section?).

Lines 402-409 I’m very much agreeing with the conclusion, yet the conclusion is not supported by the last lines of the abstract. Please rephrase the ending of the abstract.

Lines 413-414 do authors think this is only a data quality aspect or perhaps also the result of a growth model with too many “free/ unknown” parameters?

Figure 1 which m0, mmax and q are used for the two lines? Do the lines differ in these parameters?

Figure 5 what is the selection criterion from Knight 1968? Is this explained in the method?

---

## Round 0.2 · Minor Revisions

The final reviewer comments need to be addressed before the paper can be accepted.

Reviewer 1 ·

Basic reporting

English is clear -- and revisions improved the paper

Experimental design

the question is well defined -- and revisions add clarity --- no issues

Validity of the findings

findings are valid -- no issues -- conclusions stated well --

Additional comments

the revisions have added clarity to the paper -- less potential reader misunderstandings with the revisions. --ready to be published -

Reviewer 2 ·

Basic reporting

no comment

Experimental design

no comment

Validity of the findings

no comment

Additional comments

The authors did a good job in revising the manuscript. Below some minor comments to improve clarity.

Line 35 maybe rephrase to “and applications in ecology, e.g. understanding outbreak dynamics (Pratchett 2005).

Lines 62-69 This is a bit confusing as the first sentence is about mass growth, while the rest relates to length growth. As authors state later, typically length growth with exponent a=0 is equal to mass growth with exponent a=2/3. Hence, “a=0” at the start of line 63 is not equal to the second “a=0” in the same line. Maybe call the length exponent aL.

Line 64 Note that Koch et al 2015 is about shellfish. Please change reference or sentence.

198-208 not fully clear when these tests are used in the analysis, e.g. what are location parameters and where are they compared?

Lines 218-243 could be moved to discussion

Lines 253-254 maybe include, “resulting in 100 models for each dataset” as authors refer in line 304 to “the 100 models”.

Line 442 change “here” to “there”

Line 450 I suggest to rephrase “model (1)” to be understandable for someone who only reads the conclusion paragraphs

Line 452 perhaps add: as the biological meaning of the model exponent, and the variation in the exponent, is not clear.

Line 453 Authors could discuss the consequences of their findings a bit further (perhaps in relation to fisheries management). What do the findings imply in relation to the widely used estimated growth parameters in FishBase and the approximately 25,000 other papers that modeled fish growth?

More a matter of style, but I would rephrase full sentences in round brackets, e.g. line 212-213,310-311, 425-426

Figure 2: maybe include mmax and minfl in fig2a

Figure 9, the example does not show that the combination of male and female data leads to “higher residuals making refutations more difficult and increasing FNR” (line 417-418), right? Yet, it shows there are large differences between males and females. So maybe move reference of figure 9 to line 416-417 “In case that these groups had a different growth pattern (different optimal exponents) (Figure 9)”

Figure 9 include y-axis label

---

## Round 0.3 · accepted · Accept

The authors have adequately addressed the final comments.